# Rapid and predictable genome evolution across three hybrid ant populations

**Pierre Nouhaud**[1]*, **Simon H. Martin**[2], **Beatriz Portinha**[1,3], **Vitor C. Sousa**[3], **Jonna Kulmuni**[1,4]*

**1** Organismal & Evolutionary Biology Research Programme, University of Helsinki, Helsinki, Finland, **2** Institute of Evolutionary Biology, University of Edinburgh, Edinburgh, United Kingdom, **3** cE3c, Centre for Ecology, Evolution and Environmental Changes, Department of Animal Biology, Faculdade de Ciências, Universidade de Lisboa, Campo Grande, Lisboa, Portugal, **4** Tvärminne Zoological Station, University of Helsinki, Hanko, Finland

* pierr3.nouhaud@gmail.com (PN); jonna.kulmuni@helsinki.fi (JK)

## Abstract

Hybridization is frequent in the wild but it is unclear when admixture events lead to predictable outcomes and if so, at what timescale. We show that selection led to correlated sorting of genetic variation rapidly after admixture in 3 hybrid *Formica aquilonia* × *F. polyctena* ant populations. Removal of ancestry from the species with the lowest effective population size happened in all populations, consistent with purging of deleterious load. This process was modulated by recombination rate variation and the density of functional sites. Moreover, haplotypes with signatures of positive selection in either species were more likely to fix in hybrids. These mechanisms led to mosaic genomes with comparable ancestry proportions. Our work demonstrates predictable evolution over short timescales after admixture in nature.

## Introduction

Hybridization is widespread and has shaped the genomes of many extant species, representing a major source of evolutionary novelties [1–3]. Understanding the evolution of hybrid genomes is important because it can shed light on how species barriers become established, on the fitness costs (e.g., incompatibilities) and benefits (e.g., heterosis) of hybridization, and help us better understand the function of genes and their interactions [4]. Variation in local ancestry patterns along hybrid genomes has been found across many taxa, including sunflowers [5], monkeyflowers [6], humans [7], swordtail fish [8,9], sparrows [10], butterflies [11,12], and maize [13]. Such variation in local ancestry reflects the interaction of recombination with neutral (e.g., drift, migration) and selective processes. Selection may lead to the fixation (adaptive introgression [1]) or purging of one ancestry component (incompatibilities; genetic load in one hybridizing species [14,15]). Recombination rate variation can modulate the effects of selection, for example, by enabling faster purging of deleterious alleles in low-recombining regions [14]. Admixture landscapes are also impacted by past demography and stochastic events, such as bottlenecks or initial admixture proportions [16]. These mechanisms can lead

**Data Availability Statement:** All FASTQ files are available on ENA under project PRJEB55288 (hybrid samples) and PRJEB51899 (F. aquilonia & F. polyctena samples). VCF files, FASTSIMCOAL2 files and scripts, and statistics computed over

genomic windows are available from figshare: https://doi.org/10.6084/m9.figshare.c.6140793.v3. Bioinformatic and MSPRIME scripts are available from https://github.com/pi3rrr3/antmixture.

**Funding:** This work was supported by Academy of Finland (www.aka.fi) no. 328961 and HiLIFE (www2.helsinki.fi/en/helsinki-institute-of-life-science) grants to JK. SHM was supported by a Royal Society University Research Fellowship URF \R1\180682 (www.royalsociety.org). VCS was supported by Fundação Ciência e Tecnologia CEECINST/00032/2018/CP1523/CT0008 and UIDB/00329/2020 grants (www.fct.pt). The funders had no role in study design, data collection and analysis, decision to publish, or preparation of the manuscript.

**Competing interests:** The authors have declared that no competing interests exist.

**Abbreviations:** AIC, Akaike information criterion; CDS, coding sequence; IO, independent origins; MAC, minor allele count; MAF, minor allele frequency; PCA, principal component analysis; SDS, sodium dodecyl sulfate; SFS, site-frequency spectrum; SNP, single-nucleotide polymorphism; SO, single origin; TMRCA, time to the most recent common ancestor.

to the fixation or near-fixation of one ancestry component at a given locus within a hybrid population, a process we refer to as sorting of genetic variation (genome stabilization [5,12,17]). A few previous studies have investigated the interplay of different neutral and selective factors across multiple admixture events, revealing predictable sorting of ancestral variation in replicated hybrid populations [9,10,12]. However, while theory predicts that the efficiency of selection on introgressed variation will quickly decrease [7,15,18], the timescale of sorting after admixture in the wild is still unclear (but see [19]).

Here, we took advantage of multiple hybrid populations between the 2 wood ant species *Formica aquilonia* and *F. polyctena* to measure how rapid and predictable the evolution of admixed genomes is in the wild and identify the key factors that determine this predictability. These 2 species are polygynous, with up to several hundreds of queens per nest. A population is a supercolony, with dozens of interconnected nests and low relatedness between individuals [20]. Although differentiation between nests within a population is low in polygynous and supercolonial species, differentiation between populations is high, likely reflecting budding as the main dispersal mode (i.e., dispersing by foot and building a new nest in the vicinity of an already established nest, reviewed in [21]). Long-distance dispersal happens via temporary social parasitism, where a single mated queen (or possibly few) enters the nest of an unrelated species, executes the local queen, and uses the local workforce to raise her first brood [21]. Several hybrid *F. aquilonia × polyctena* populations with distinct mitochondrial sequences have been previously characterized in Southern Finland [22], providing a test case for the outcomes of admixture in nature.

## Results and discussion

We generated whole-genome sequence data from 3 hybrid populations (Fig 1A, *n* = 39) and used genomes from both species collected within and outside their overlapping range (*n* = 10 per species [23]; mean coverage: 20.6×, S1 Table). Analyzing ca. 1.6 million single-nucleotide polymorphisms (SNPs) genome-wide, we confirmed that the hybrid populations were genetically intermediate between *F. aquilonia* and *F. polyctena* (Figs 1B and 1C and S1 and S1 Table). Both Bunkkeri and Pikkala individuals carried distinct *F. aquilonia*-like mitotypes (Fig 1D). *F. polyctena*-like mitotypes were observed in the Långholmen population (Fig 1D), where 2 hybrid lineages termed W and R coexist (S1 Fig, [23]). These lineages basically share a single mitotype and are possibly maintained through environment-dependent genetic incompatibilities and assortative mating [24,25]. The 3 hybrid populations have highly differentiated mitotypes (108 mutational steps between both *F. aquilonia*-like and *F. polyctena*-like clusters, Fig 1D) and nuclear DNA (mean pairwise $F_{ST}$ estimates between hybrid populations ranging from 0.18 to 0.23) but low diversity of mitotypes within a population (≤2). These results are consistent with population bottlenecks during colony establishment coupled with little long-distance dispersal, as described above.

To elucidate the ancestry of the hybrid populations and date admixture events, we reconstructed the demographic histories of pairs of hybrid populations using a coalescent approach based on the site-frequency spectrum (SFS) of nuclear SNPs (FASTSIMCOAL2 [26]). Coalescent analyses support balanced admixture proportions between species (i.e., no apparent minor ancestry; Fig 1E and S2–S11 Tables), with comparable parameter estimates under scenarios assuming a single origin (SO, 1 admixture event) or independent origins (IO, multiple admixture events, S7–S11 Tables). Consistent with field observations, assuming hybridization events occurred over the last 50 generations led to higher likelihoods (compared to older admixture events), with admixture time estimates ranging from 14 to 47 generations ago (S2–S11 Tables). Model choice provided more support towards an IO

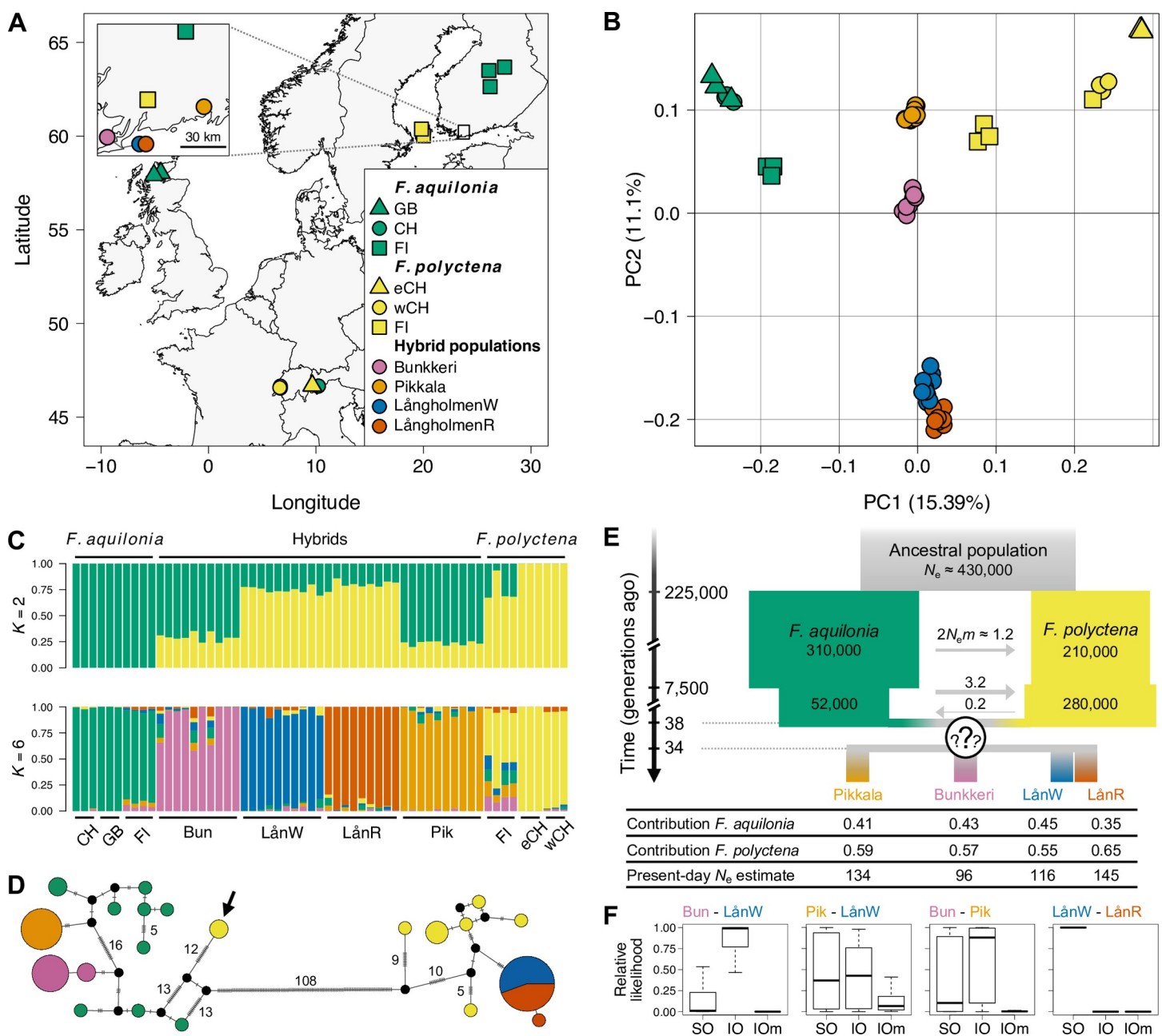

**Fig 1. Young, independently evolving hybrid wood ant populations between *F. aquilonia* and *F. polyctena* in Southern Finland. (A)** Sampling sites across Europe (eCH: East Switzerland, wCH: West Switzerland, map base layer from Natural Earth: https://www.naturalearthdata.com/downloads/110m-physical-vectors). **(B)** Principal component analysis of 46,886 nuclear SNPs (5 kbp-thinned, MAC ≥ 2). PC1 discriminates between both species and PC2 between hybrid populations. Colors and symbols as in panel A. **(C)** sNMF estimation of individual ancestry coefficients for $K = 2$ and $K = 6$ populations (cross-entropy criterion gives $K = 6$ as the best $K$, see S1 Table for detailed admixture proportions computed using sNMF$_{K = 2}$, LOTER and naive chromosome painting outputs). **(D)** Haplotype network derived from 199 mitochondrial SNPs. Circles represent haplotypes, with sizes proportional to their counts. Dashes indicate the number of mutational steps, with numbers ≥5 provided. The black arrow indicates 2 Finnish *F. polyctena* individuals carrying *F. aquilonia*-like mitotypes. **(E)** Admixture history between *F. aquilonia* and *F. polyctena* inferred through an SFS-based approach. The question marks represent the uncertainty associated with the admixture model: parameter estimates are comparable under both single origin and independent origins scenarios, but single origin models support separation of hybrid populations after brief periods of shared ancestry. $N_e$: average effective population size in number of haploids, $m$: migration rate. **(F)** Results of the model choice analysis performed with FASTSIMCOAL2 for each population pair. SO: single origin scenario; IO: independent origins scenario; IOm: independent origins scenario with migration between hybrid populations after admixture. The data underlying this figure can be found in https://doi.org/10.6084/m9.figshare.c.6140793.v3. MAC, minor allele count; SFS, site-frequency spectrum; SNP, single-nucleotide polymorphism.

scenario for the Bunkkeri—LångholmenW pair (median relative likelihoods: $L_{IO}$ = 0.99, $L_{SO}$ = 0.01; Fig 1F) and Bunkkeri—Pikkala ($L_{IO}$ = 0.88, $L_{SO}$ = 0.10; Fig 1F) and towards an SO scenario for the LångholmenW—LångholmenR pair ($L_{IO}$ = 0, $L_{SO}$ = 1.00; Fig 1F). Results were inconclusive for the remaining pair (Pikkala—LångholmenW: $L_{IO}$ = 0.43, $L_{SO}$ = 0.37; Fig 1F). Parameter estimates from models that assume SO indicate that even if the hybrid populations originate from a single admixture event, they mostly evolved independently (on average 9.5 generations of shared ancestry since admixture, S7–S11 Tables). Following the admixture events, no significant gene flow was inferred either between hybrid populations or between hybrids and both species ($L_{IOm}$ < 0.12 in all comparisons, S7–S11 Tables and Fig 1F). These demographic reconstruction results and patterns of mitochondrial variation are consistent with 2 alternative scenarios for the origin of these hybrid populations. Either they arose through independent hybridization events or an ancestral hybrid population combining several matrilines from both species was established and split into 3 locations, spanning 60 km within <50 generations. Considering wood ant reproductive and dispersal biology, we suggest independent admixture events (IO) as a more likely scenario, in line with model choice supporting IO for 2 population comparisons. However, we acknowledge that reconstructing very recent events accurately is challenging, and we next evaluate our results in the light of both IO and SO scenarios.

To investigate how evolution has shaped hybrid genomes after admixture, we mapped ancestry components along chromosomes independently for each hybrid population. To do this, we inferred local ancestry at 1.5 million phased SNPs using LOTER (Fig 2A [27]) and quantified tree topology weights in 100-SNP windows with TWISST (Fig 2B [28]). Hybrid populations have strongly correlated admixture landscapes along the genome (i.e., local ancestry in 1 population predicts the local ancestry in another population, Figs 2D and S3, Spearman's rank correlation coefficients computed from TWISST weights ranging from 0.51 to 0.62, $P < 10^{-15}$ for all population pairs). To test whether such predictability would be expected under neutrality, we used MSPRIME [29] to simulate neutral admixture events following both SO and IO scenarios for each hybrid population pair, using demographic parameters inferred with FASTSIMCOAL2 for the same pair. In all instances (4 population pairs × 2 admixture scenarios), neutral simulations led to balanced contributions of both ancestry components along the genome, but did not capture the clear deviations towards either ancestry component observed locally in the genome (i.e., sorting) with our empirical data (Fig 2C–2E, two-sample Kolmogorov–Smirnov tests, $P < 10^{-15}$ for all populations, S3 and S4 Figs and S12 and S13 Tables). Thus, the extent of correlated sorting among hybrid populations cannot be explained solely by neutral processes, including the admixture scenario (SO versus IO) and/or demographic history. Moreover, both SFS-based demographic modeling (Fig 1F) and the lack of mitochondrial haplotype sharing between hybrid populations (Fig 1D) argue against gene flow as a potential source for the parallelism observed. As such, other mechanisms must be invoked to explain the rapid evolution of sorted and correlated admixture landscapes in the different hybrid populations.

We hypothesized that correlated sorting of genetic variation in hybrid populations is caused by selection against deleterious alleles that have accumulated in the hybridizing species with the lowest effective population size ($N_e$ [7,8,30,31]). This effect is expected to be stronger in gene-dense regions [8,32] but also in low-recombining regions [14], where tighter linkage between deleterious alleles, and between neutral and deleterious alleles, leads to removal of larger tracts of ancestry. The 2 *Formica* species were estimated to have contrasting effective population sizes, with a ca. 30% lower $N_e$ in *F. polyctena* compared to *F. aquilonia* in the last 200,000 generations ([23], Fig 1E). In hybrid populations, sorting (hereafter ≥90% of either ancestry component inferred from LOTER at a given locus) was faster in low-recombining

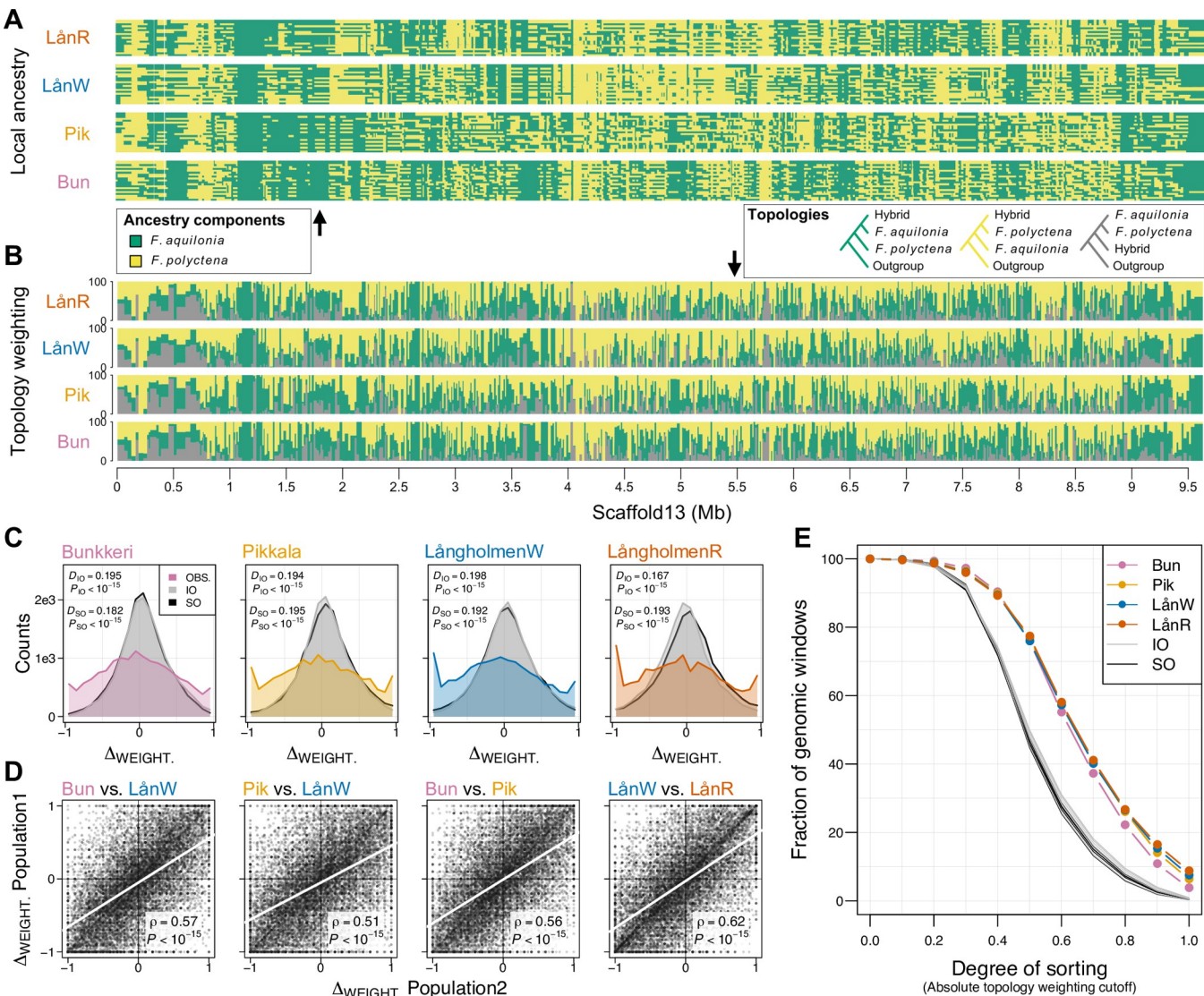

**Fig 2. Sorting of genetic variation is more correlated than expected under neutrality across 3 hybrid wood ant populations.** Examples of **(A)** local ancestry and **(B)** topology weighting patterns (green: hybrids locally related to *F. aquilonia*, yellow: hybrids locally related to *F. polyctena*) inferred independently in each population along 1 pseudo-chromosome (Scaffold 13). **(C)** Excess of extreme topology weightings in observed compared to simulated data (IO, SO) in all populations. Empirical and simulated distributions compared with two-sample Kolmogorov–Smirnov tests ($D$, test statistic and $P$, $P$-value). **(D)** Genome-wide comparison of topology weighting differences between each population pair ($\Delta_{\text{WEIGHT.}}$: *F. aquilonia* weighting minus *F. polyctena* weighting, computed per population over 14,890 100-SNP windows, gray circles). The regression line is indicated in white. ρ, Spearman's correlation coefficient and $P$, $P$-value of the Spearman's correlation test. **(E)** A larger fraction of the genome is sorted in observed compared to simulated data in all populations (sorting measured as the absolute *F. aquilonia* or *F. polyctena* weighting, see S3 Fig for detailed results per pairwise comparison). The data underlying this figure can be found in https://doi.org/10.6084/m9.figshare.c.6140793.v3. IO, independent origins; SO, single origin.

regions of the genome, as well as in gene-rich regions (Figs 3A and S5). Moreover, in low-recombining regions, the *F. aquilonia* ancestry was preferentially fixed in all populations (Fig 3A). Focusing on coding SNPs, we found a significant enrichment for *F. aquilonia* ancestry genome-wide in all populations, consistent with the hypothesis that hybrid populations have purged the deleterious load accumulated in *F. polyctena* due to its smaller $N_e$ (genomic permutations, $P < 0.002$ in all populations, Fig 3B). These results support the contributions of both recombination rate variation and genetic load in promoting sorting of ancestral variation in

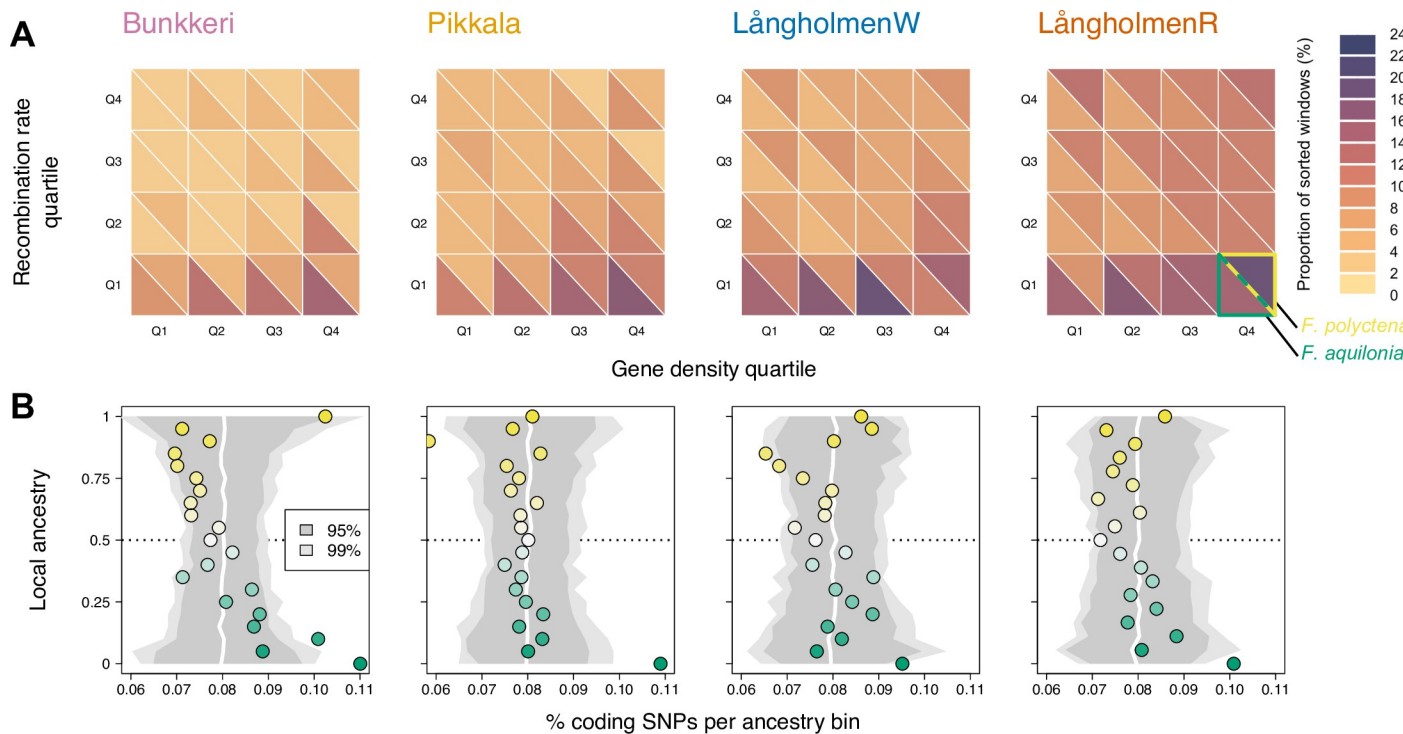

**Fig 3. Sorting of ancestral polymorphism in hybrids is driven by recombination rate variation and genetic load.** (A) Heatmap showing the fraction of sorted 20 kbp windows and the direction of sorting (ancestry component fixed) as a function of recombination rate and gene density quantiles in each hybrid population. (B) Coding regions are significantly enriched for the *F. aquilonia* ancestry component in all hybrid populations ($P < 0.002$). For each population (panel, same as A) is plotted local ancestry (y-axis, 0: *F. aquilonia* allele fixed, 1: *F. polyctena* allele fixed) as a function of the fraction of SNPs within CDS (x-axis). Confidence intervals (in gray) were obtained using 500 genomic permutations (white line: median of the permutation approach). The data underlying this figure can be found in https://doi.org/10.6084/m9.figshare.c.6140793.v3. CDS, coding sequence; SNP, single-nucleotide polymorphism.

hybrids, as previously characterized in other study systems (reviewed in [33]). Our study also reveals that consistent sorting of ancestral variation can happen in less than 50 generations in small populations (Fig 1E).

Positive selection could contribute to the correlated sorting of genetic variation across hybrid populations: advantageous alleles from either species could repeatedly sweep in distinct hybrid populations after admixture. Under this scenario, a genomic region fixed for the *F. aquilonia* ancestry component in hybrids would display signatures of selection in *F. aquilonia* individuals. To test this hypothesis, we looked for selective sweep signatures in both hybridizing species with RAiSD [34], which quantifies changes in the SFS, levels of linkage disequilibrium and genetic diversity along the genome through the composite sweep statistic μ (Fig 4). Consistently sorted genomic windows (i.e., windows fixed for either species ancestry across all hybrid populations, 1.92% of the windows overall) displayed significantly higher sweep statistics only in the species from which the ancestry component was fixed in hybrids (genomic permutations, $P < 0.001$, Fig 4B). While recombination rate estimates were significantly lower than the rest of the genome in these consistently sorted windows (Wilcoxon test, $W = 1,740,660, P < 10^{-15}$), purging of load in low-recombining regions cannot explain the observation that hybrids have fixed ancestry from the species where a sweep may have occurred prior to hybridization (Fig 4B).

Hybrid genomes provide powerful insights into evolution because they are exposed to strong, and often opposing, selective forces [33]. In this study, we coupled reconstruction of admixture histories, local ancestry inference, and coalescent simulations to show that the

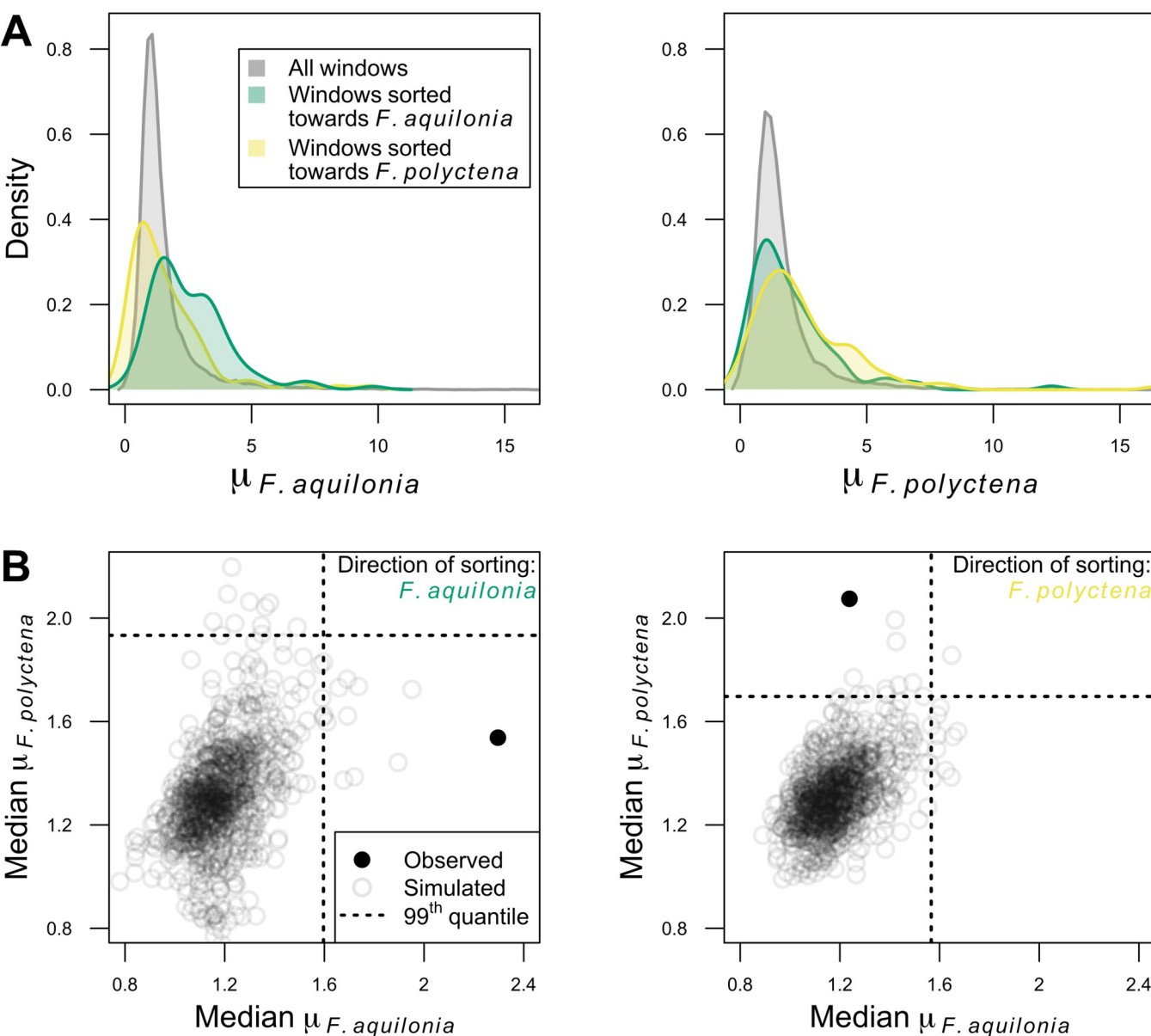

**Fig 4. Signatures of selective sweeps in hybridizing species predict the direction of sorting in admixed genomes. (A)** Distribution of selective sweep statistics (μ) computed over 20 kbp windows in *F. aquilonia* (left) and *F. polyctena* (right). Genome-wide μ distribution (gray) and observed values in windows fixed for either *F. aquilonia* ($n$ = 104 windows, green) or *F. polyctena* ancestry components ($n$ = 98, yellow) in all hybrid populations. **(B)** Windows fixed for either *F. aquilonia* (left) or *F. polyctena* (right) ancestry components in all hybrid populations are significantly enriched for high μ values in *F. aquilonia* or *F. polyctena* individuals, respectively. This suggests that a haplotype with a signature of positive selection in either species is more likely to fix in hybrids. Simulated μ values were obtained through 1,000 genomic permutations (as in [64]). Each circle represents medians computed over all consistently sorted windows (solid: observed, open: simulated). The data underlying this figure can be found in https://doi.org/10.6084/m9.figshare.c.6140793.v3.

sorting of ancestral variation is predictable and, in some instances, likely independent across several natural hybrid ant populations. Some predictability has been previously characterized in other systems (e.g., [9]), and introgression is for example limited on sex chromosomes compared to autosomes in replicated hybrid populations of both Italian sparrows [10] and *Lycaeides* butterflies [12]. We also documented that the known interplay between negative selection and recombination rate variation contributes to remarkable correlation of ancestry components along the genome between hybrid wood ant populations (Spearman's rank correlation

coefficients using TWISST ranging from 0.51 to 0.62 in hybrid wood ants, Fig 2D). Since ancestry proportions are balanced in hybrid wood ants, negative selection should not target any minor ancestry component, as assumed under unbalanced ancestry proportions and when species barriers are highly polygenic (e.g., [8]). Instead, our results suggest that negative selection is impacting ancestry from the species with the smaller effective population size, and presumably a higher load of deleterious alleles. Distinguishing signatures of incompatibilities from those of genetic load and their possible interplay remains a challenge for future studies.

We also showed, to our knowledge for the first time, that events of positive selection prior to admixture likely contribute to the predictability of admixture outcomes (see [35] for a theoretical treatment): Genomic regions displaying signatures of selective sweeps in 1 hybridizing species tend to fix the same ancestry component in hybrid populations. These genomic regions could also act as incompatibilities, which we cannot identify on the sole basis of our data, but which impact the landscape of introgression in hybrids [9], as previously documented in our study system [23,36]. In the future, novel methodological developments [37,38] coupled with larger sample sizes may allow identifying candidate incompatibilities in hybrid wood ants.

Finally, in contrast to other recent studies of hybrid genome evolution, the ant hybrids still show balanced ancestry contributions after ca. 50 generations since admixture. Fluctuating, environment-dependent selection could be one mechanism maintaining both ancestry components in hybrids, as microsatellite allele frequencies of the cold-adapted *F. aquilonia* species have been shown to positively correlate with yearly temperature over a 16-year time period in one of the hybrid populations we studied [25]. As this correlation was stronger in males, haplodiploidy is another mechanism that may contribute to the maintenance of genetic variation in wood ants. To conclude, we have shown that the sorting of ancestral genetic variation in hybrid genomes can occur rapidly and predictably after admixture due to both positive and purifying selection.

## Methods

### Sampling

*F. aquilonia*, *F. polyctena*, and their Finnish hybrids are polygynous: within a nest, the reproductive effort is shared across dozens or hundreds of egg-laying queens. These 2 species and their Finnish hybrids are also supercolonial, with populations (i.e., supercolonies) formed by the association of several cooperating nests within a site. Polygyny and supercoloniality both result in low relatedness among individuals sampled within a given population [39]. We sampled hybrid individuals from 3 populations previously mapped in Southern Finland: Långholmen (composed of 2 hybrid lineages R & W; [23]), Bunkkeri and Pikkala [22]. We collected unmated queens from Bunkkeri ($n = 10$) and Långholmen ($n_W = 10$, $n_R = 9$) in Spring 2018 and workers from Pikkala ($n = 10$) in Spring 2015 (S1 Table). Data generated by Portinha and colleagues [40] were used as *F. aquilonia* and *F. polyctena* reference panels. Briefly, it consists of 10 workers (diploid females) per species sampled from several monospecific colonies across Europe (Fig 1A and S1 Table). *F. polyctena* samples were collected at 2 locations in Switzerland (East, $n = 3$; and West, $n = 3$), the Åland islands (Finland, $n = 3$) and Southern Finland ($n = 1$), and *F. aquilonia* samples were collected in Scotland (UK, $n = 3$), East Switzerland ($n = 3$), Central Finland ($n = 3$), and Southern Finland ($n = 1$). Both species are sympatric in Southern Finland, while they can be considered allopatric in other sampling locations (they are found at different altitudes in East Switzerland [40]).

### DNA extraction

Both the hybrid samples generated for this study and the *F. aquilonia* and *F. polyctena* samples from Portinha and colleagues [40] were processed and sequenced at the same time, and all

data went through the same pipeline. DNA was extracted with a sodium dodecyl sulfate (SDS) protocol from whole bodies, and sequencing libraries were built with NEBNext DNA Library Prep Kits (New England Biolabs) by Novogene (Hong Kong).

### DNA sequencing and read mapping

Unless stated otherwise, all software was used with default parameters. Whole-genome sequencing was carried out on Illumina Novaseq 6000 (150 base pairs, paired-end reads), targeting 15× per individual (S1 Table). We trimmed raw Illumina reads and adapter sequences with TRIMMOMATIC v0.38 and mapped trimmed reads against the *F. aquilonia* × *F. polyctena* reference genome [41] using BWA MEM v0.7.17 [41]. We then removed duplicates using PICARD TOOLS v2.21.4 (http://broadinstitute.github.io/picard). All bioinformatic scripts are available from https://github.com/pi3rrr3/antmixture.

### SNP calling and filtering

We called SNPs jointly across all samples with FREEBAYES v1.3.1 (population priors disabled with -k option [42]) and normalized the resulting VCF file (parsimonious left-alignment of multi-nucleotide variants) using VT v0.5 [43]. We excluded both sites located at less than 2 base pairs from indels and sites supported by only forward or reverse reads using BCFTOOLS v1.10 [44]. We decomposed multi-nucleotide variants using vcfallelicprimitives from VCFLIB v1.0.1. The next steps were carried out with BCFTOOLS. Biallelic SNPs with quality equal or higher than 30 were kept. Individual genotypes with (i) genotype qualities lower than 30 and/ or (ii) with depth of coverage lower than 8 were coded as missing data. Sites displaying more than 50% missing data over all samples were discarded. Genotyping errors due to, e.g., misaligned reads were removed using a filter based on excessive heterozygosity. To do so, we used an approach similar to Pfeifer and colleagues [45] and pooled all samples together, after which we excluded sites displaying heterozygote excess ($P < 0.01$,—hardy command from VCFTOOLS v0.1.16 [46]). Since putative genetic incompatibilities should not be heterozygous in hybridizing *F. aquilonia* and *F. polyctena* individuals, they should not be impacted by this filtering step. Overall, 122,044 sites were removed, half of them located on unanchored, repeat-rich contigs and displaying heterozygosity ≥0.48 in all populations. We then filtered sites based on individual sequencing depth distributions at SNP loci, setting as missing sites where depth was lower than half or higher than twice the mean value of the individual considered. Finally, sites with more than 15% missing data over all samples were discarded. These steps led to a final dataset of 1,659,532 SNPs across 59 individuals.

### Population structure

Population structure was assessed using a reduced dataset of 46,896 SNPs obtained after thinning (retaining 1 SNP every 5 kbp) and discarding sites with minor allele count (MAC) <2. We performed a principal component analysis (PCA) with PLINK v1.9 [47] and sNMF clustering using the LEA package v3.0.0 [48] in R v3.6.2 [49]. Clustering was carried for a number of ancestral components ($K$) ranging from 1 to 10, with 10 iterations performed per $K$ value. The lowest cross-entropy was obtained with $K = 6$, and the results of runs with the lowest cross-entropy for both $K = 2$ and $K = 6$ are shown in Fig 1C.

### Mitotype network

Mitochondrial SNPs were called separately with FREEBAYES using a frequency-based approach (—pooled-continuous option). SNP filtering was carried using the same pipeline as

for the nuclear genome, which led to the identification of 199 biallelic SNPs across 59 individuals. Individual FASTA files were written using vcf-consensus and aligned with MAFFT v7.429 [50]. The median-joining network was created using POPART [51].

## Demographic modeling

Before reconstructing admixture histories, we removed the 122,479 SNPs located on the third chromosome. This chromosome carries a supergene controlling whether *Formica* colonies are headed by 1 or multiple queens (social chromosome [52]). Recombination reduction between the 2 supergene variants leads to the maintenance of ancestral polymorphisms across *Formica* species that could bias our demographic inference. The dataset used for demographic modeling hence comprises 1,537,053 SNPs.

We used the composite-likelihood method implemented in FASTSIMCOAL2 v2.6 [26] to compare alternative demographic models to demographic parameters inferred from the SFS following Portinha and colleagues [40]. We ran each model 100 times with 80 iterations per run for likelihood maximization and the expected SFS was approximated through 200,000 coalescent simulations per iteration. We assumed a mutation rate of $3.5 \times 10^{-9}$ per bp per haploid genome per generation, which is an average based on estimates currently available for social insects [53]. No population growth was allowed, but population sizes could change when migration rates changed. Generation time was assumed to be 2.5 years [40]. Finally, we used the speciation history inferred by Portinha and colleagues [40] to constrain parameter range prior to the admixture event(s) in our demographic models. This speciation history was inferred from Finnish *F. aquilonia* and *F. polyctena* individuals including those used in the present study. All parameter ranges are indicated in S2 Table (three-population models) and S7 Table (four-population models).

Field observations suggest hybrid populations may have arisen through recent admixture (ca. 50 years ago). Constraining admixture under 50 generations (ca. 125 years ago) led to models with higher expected likelihoods for all hybrid populations, and both constrained and unconstrained results are shown in S3–S6 Tables.

## SFS characteristics

We built folded SFSs using minor allele frequency (MAF) and downsampled genotypes to minimize missing data, using R scripts available at https://github.com/vsousa/EG_cE3c/tree/master/CustomScripts/Fastsimcoal_Example_Bootstrap/Scripts_VCFtoSFS. We first determined a minimum sample size across all sites (number of individuals available for resampling minus maximum number of missing data per site). We then resampled individuals in 50 kbp windows and discarded blocks where the mean distance between 2 consecutive SNPs within a block was <2 bp. We estimated the number of monomorphic sites from the proportion of polymorphic sites and the total number of callable sites. The latter was obtained from each individual BAM file using MOSDEPTH v0.2.9 [54] and individual sequencing depth thresholds defined for SNP calling.

Distinct 3D- and 4D-SFSs were built for our three- and four-population models, respectively, to answer specific study questions (see below). In both cases, we used the single individual from each species sampled in Southern Finland as representative of their respective species. For each hybrid population, we resampled 4 individuals every window to build the SFSs. The 3D-SFSs contained information of both species and 1 focal hybrid population, while the 4D-SFSs included information of both species and 2 focal hybrid populations. For these latter models, we analyzed 4 different pairwise combinations: Bunkkeri—LångholmenW, Pikkala—LångholmenW, Bunkkeri–Pikkala, and finally LångholmenW—LångholmenR.

## Disentangling between secondary contact and hybridization: Three-population models

For each hybrid population, we tested 3 different scenarios that could lead to present-day admixed individuals. The first scenario is hybridization, namely an admixture event between *F. polyctena* and *F. aquilonia* where one species would contribute a genetic input of $\alpha$ into the hybrid population, while the other species would contribute the remaining fraction $1 - \alpha$. This scenario was assessed both with and without gene flow between hybrids and either (or both) species after admixture (forward in time).

The second scenario is secondary contact, where after the speciation event hybrid ancestors would first diverge from one species, and then receive (haploid) migrants from the other species. This scenario was tested in both directions (i.e., assuming both ((*F. polyctena*, hybrid), *F. aquilonia*) and ((*F. aquilonia*, hybrid), *F. polyctena*) topologies). Gene flow was also allowed between both species before and after the split between the hybrid population and the first species.

The third and last scenario is a trifurcation model, where the 2 species and the hybrid ancestral population would first diverge simultaneously, after which all populations would exchange migrants at different rates. Higher migration from both species into the hybrid ancestral population would lead to admixed individuals in the current-day hybrid population.

## Disentangling between single and independent origins of hybrid populations: Four-population models

Hybrid populations that arose through a single admixture event followed by a long period of shared ancestry would have more correlated sorting of genetic variation than if they separated soon after the admixture event or arose through independent admixture events. To disentangle between these scenarios, we tested 2 alternative admixture models using 2 hybrid populations at a time. Based on the results of our three-population models, admixture times were constrained to the last 50 generations in all subsequent models.

The first model is an SO scenario where *F. polyctena* contributes a proportion $\alpha$ of the genetic material of the ancestral hybrid population, with *F. aquilonia* providing the complementary $1 - \alpha$ fraction. This ancestral hybrid population then diverges into 2 hybrid populations.

The second model is an IO scenario where each hybrid population arises through a distinct admixture event, with possibly different contributions from both species (i.e., *F. polyctena* contributes a fraction $\alpha$ for the first hybrid population and $\beta$ for the second hybrid population, and *F. aquilonia* $1 - \alpha$ and $1 - \beta$, respectively). As post-admixture gene flow between hybrid populations could also lead to correlated sorting of genetic variation, we additionally tested an IO with migration (IOm) scenario that includes reciprocal migration between 2 hybrid populations after the most recent admixture event.

## Model choice

We performed model choice using relative likelihoods computed based on Akaike information criterion (AIC) to disentangle between SO, IO, and IOm scenarios for each hybrid population pair, following Excoffier and colleagues [55]. To minimize the impact of linkage, observed SFSs were built as described previously for parameter estimation, but using a pruned dataset, keeping every 100th SNP (18,378 SNPs in total) after filtering sites where at least 4 genotypes were available per hybrid population. We computed the AIC for 100 bootstrap replicates, resampling individuals for each SNP, using R scripts available at https://github.com/vsousa/

EG_cE3c/tree/master/CustomScripts/Fastsimcoal_Example_Bootstrap/Scripts_VCFtoSFS.
For each bootstrap replicate, likelihoods were computed based on average expected SFSs simulated 100 times using the maximum-likelihood estimates of each model, with 200,000 coalescent simulations run per replicate. Both IO and SO models had 8 parameters, while the IOm model had 10.

### Population recombination rate estimation

We used iSMC v0.0.23 [56] to estimate population recombination rates along the genome. We hypothesized that the recombination landscape in hybrids would be an average of the recombination landscapes in both species. Using all non-Finnish individuals from both species jointly, we fitted a model of coalescence with recombination including 40 TMRCA (time to the most recent common ancestor) intervals and 10 $\rho$ categories. Population recombination rate estimates were collected over non-overlapping 20 kbp windows, discarding windows with less than 20 SNPs.

### Haplotype estimation

Prior to mapping ancestry components, we phased all SNPs anchored on scaffolds (78.2% of the genome) with WHATSHAP v1.0 [57], which uses short-range information contained within paired-end reads. We then performed statistical phasing and imputation with SHAPEIT v4.1.2 [58], using the sequencing data setting and increasing the MCMC iteration scheme as indicated in SHAPEIT documentation. The total phased dataset contained 1,490,364 SNPs.

### Outgroup information

We used *F. exsecta* as an outgroup to root topologies inferred with TWISST (see below). This species belongs to a distinct species group (*F. exsecta* group) while *F. aquilonia* and *F. polyctena* both belong to the *F. rufa* group [59]. To extract *F. exsecta* genotypes at our phased SNP loci, we mapped data previously generated by Dhaygude and colleagues [60] against the same reference genome used in the present study. These data consist of Illumina paired-end, $2 \times 100$ bp reads generated from a pool of 50 (haploid) males (9.89 Gbp overall, median insert size: 469 bp, ENA accession SAMN07344806). Reads were trimmed with TRIMMOMATIC using the same parameters as before, mapped with BWA MEM, and deduplicated with PICARD TOOLS. We filtered proper read pairs with mapping quality $\geq$20 (86.4% of the reads, sequencing depth 14.5×) using SAMTOOLS v1.13 [61]. We generated a pileup file (disabling per-base alignment quality computation and filtering minimum base quality $\geq$20) and finally extracted *F. exsecta* genotypes at each locus of the phased dataset using a custom R script.

### Mapping ancestry components along the genome

We used 3 different methods to map ancestry variations along the genome, which all relied on reference panels comprising both *F. aquilonia* and *F. polyctena* individuals. As gene flow from *F. aquilonia* to *F. polyctena* in Finland prior to admixture [40] could bias our results, we ran all 3 methods after excluding the Finnish representatives of both species from our reference panels (S1 Table).

We performed local ancestry inference from phased data for each population independently using LOTER v1.0 [27], after which we averaged local ancestries over the same non-overlapping 20 kbp windows used for population recombination rate estimation.

We also recorded topology weightings in 100-SNP windows along the genome with TWISST v0.2 [28] using phased data. Analyzing jointly the 2 species, the outgroup *F. exsecta* and 1 hybrid population at a time, the 3 rooted topologies are (Fig 2B):

1. (((*F. aquilonia*, hybrid), *F. polyctena*), outgroup),

2. (((*F. polyctena*, hybrid), *F. aquilonia*), outgroup),

3. (((*F. aquilonia*, *F. polyctena*), hybrid), outgroup).

Since the average weighting of the third topology was around 10% genome-wide in all hybrid populations, we measured topology weighting differences per window by subtracting the weighting of the second topology (*F. polyctena* topology) to the weighting of the first topology (*F. aquilonia* topology, Fig 2C and 2D). The resulting metric, $\Delta_{\mathrm{WEIGHT}}$, ranges from −1 to +1 (in a given window, only the *F. polyctena* or the *F. aquilonia* topologies are inferred, respectively). When $\Delta_{\mathrm{WEIGHT}}$ is close to zero, both topologies have similar weightings, which we interpreted as a lack of sorting of ancestral variation in hybrids.

Finally, we also performed a "naive" chromosome painting approach from non-phased genotypes. To do so, we first discarded sites with more than 2 missing genotypes over all reference individuals, after which we identified 79,336 ancestry-informative SNPs displaying an allele frequency difference above 80% between both species.

## Coalescent simulations

We used coalescent simulations to measure sorting levels that would be expected in each hybrid population under the reconstructed admixture history. For each population pair used for demographic reconstruction, we simulated both IO and SO scenarios (4 population pairs × 2 scenarios). We used the parameters inferred under each scenario for each population pair with FASTSIMCOAL2 (parameters: divergence and admixture times, $N_e$ estimates, size changes, migration rates, and admixture proportions) and ran simulations in MSPRIME v1.0.2 [29], modeling 100 non-recombining 10 kbp blocks with a recombination rate of $10^{-6}$ within blocks. Each simulation was run 100 times, and VCF files were produced via MSPRIME assuming a mutation rate of $3.5 \times 10^{-9}$ per bp per haploid genome per generation [53] (S12 and S13 Tables and S4 Fig). Topology weightings were computed directly from tree sequences using TWISST and $\Delta_{\mathrm{WEIGHT}}$ distributions were obtained as stated above over all 100 replicates. MSPRIME scripts are available from https://github.com/pi3rrr3/antmixture.

## Selective sweep detection

We looked for evidence for selective sweeps independently in both hybridizing species with RAiSD v2.9 [34] using the full dataset. The composite selective sweep statistics μ were estimated with default parameters using non-Finnish individuals from each species. Results were then averaged over 20 kbp non-overlapping windows in R (Fig 4A).

## Genomic permutation approach

We tested statistical significance of both (i) the enrichment in *F. aquilonia* ancestry at coding SNPs (Fig 3B) and (ii) the association between the direction of sorting and the evidence for selective sweeps in 1 species (Fig 4B) using a similar shift-based, circular permutation scheme inspired by Yassin and colleagues [62].

For the first analysis, we slid the local ancestry landscape inferred by LOTER at the SNP level by 100 kbp increments (i.e., five 20 kbp windows at a time), maintaining the structure of ancestry blocks as observed in our data. For each shift replicate, we then computed the fraction

of coding SNPs within each local ancestry bin genome-wide. For this analysis, we ran 500 per-mutations and *P*-values were defined as the proportion of shift replicates in which at least a similar fraction of coding SNPs was reached as in our empirical data. The 95th and 99th quantiles plotted in Fig 3B were computed over all 500 permutations.

For the second analysis, we slid the average local ancestry values computed over 20 kbp non-overlapping windows by 100 kbp increments, which also shifted the location of sorted windows. For each shift replicate, we then computed the median composite selective sweep statistic μ across all sorted windows genome-wide for each ancestry component independently. We ran 1,000 permutations and defined *P*-values as the proportion of shift replicates in which median μ values were at least as high as observed in the empirical data (Fig 4B).

### Density of functional sites and gene content in sorted regions

The density of functional sites was computed in 20 kbp non-overlapping windows along the genome by measuring for each window the fraction of base pairs falling within coding sequences (CDS), which positions were extracted from the GFF annotation file [63].

The 202 sorted windows (defined as displaying $\geq$ 90% of the same ancestry component across all hybrid populations, see Fig 4) intersected with 364 gene models; however, no significant gene enrichment was detected with TOPGO v1.0 [64].

### Supporting information

**S1 Fig. Visualization of the first 5 principal components of the principal component analysis performed over 46,886 SNPs genome-wide (5 kb-thinned, MAC $\geq$ 2).** The data underlying this figure can be found in https://doi.org/10.6084/m9.figshare.c.6140793.v3.
(TIF)

**S2 Fig. Comparison of ancestry mapping approaches.** For each hybrid population (columns) are shown TWISST $\Delta_{WEIGHT.}$ statistics vs. LOTER local ancestry estimates (first row, 14,890 100-SNP windows), TWISST $\Delta_{WEIGHT.}$ statistics vs. naive chromosome painting local ancestry estimates (PAINTING, second row, 5,529 windows with at least 5 ancestry-informative SNPs), and LOTER vs. naive chromosome painting local ancestry estimates (third row, 5,529 windows with at least 5 ancestry-informative SNPs). $\Delta_{WEIGHT.}$ ranges between −1 if all topologies in the window group the hybrid population with *F. aquilonia*, to +1 if with *F. polyctena*. LOTER and naive chromosome painting are both SNP-based (results averaged over windows) and code ancestries as 0 for *F. aquilonia* and 1 for *F. polyctena*. In each panel, the regression line is indicated in white. ρ, Spearman's correlation coefficient and *P*, *P*-value of the Spearman's correlation test. The data underlying this figure can be found in https://doi.org/10.6084/m9.figshare.c.6140793.v3.
(TIF)

**S3 Fig. Observed (OBS.) and simulated levels of local ancestry correlation (left, each point is a chromosome) and sorting (right) for each hybrid population pair (rows).** The degree of sorting is measured as the absolute *F. aquilonia* or *F. polyctena* weighting. IO: independent origins scenario, SO: single origin scenario (100 independent runs per scenario). The data underlying this figure can be found in https://doi.org/10.6084/m9.figshare.c.6140793.v3.
(TIF)

**S4 Fig. Principal component analyses of observed and simulated datasets for each hybrid population pair (rows).** Observed PCAs were obtained as per Fig 1 (5 kb-thinned SNP data, minor allele count $\geq$ 2). Simulations were run with msprime using parameter estimates inferred under both single and independent origins scenarios with fastsimcoal2 and assuming

a mutation rate of $3.5 \times 10^{-9}$. One run was randomly picked per simulated scenario. The data underlying this figure can be found in https://doi.org/10.6084/m9.figshare.c.6140793.v3. (TIF)

**S5 Fig.** Distribution of LOTER local ancestry estimates (x-axis, 0: fixed for *F. aquilonia* ancestry component, 1: fixed for *F. polyctena* ancestry component) across recombination rate (upper row) and gene density (lower row) quartiles in each hybrid population (columns), computed over 20 kbp non-overlapping windows. Medians are indicated with black dots. The data underlying this figure can be found in https://doi.org/10.6084/m9.figshare.c.6140793.v3. (TIF)

**S1 Table. Sample information, sequencing statistics, hybrid indices, and accession numbers for each sample analyzed in the study.** In the caste column, w: worker and q: young unmated queen. Principal component coordinates (PC1 and PC2) and *F. aquilonia* admixture proportions (sNMF clustering analysis run with $K = 2$, sNMF_K2_Faq) are both based on a 5 kbp-thinned dataset of 46,896 SNPs with MAC $\geq$ 2. HI_loter values are based on 1,490,364 phased SNPs, with non-Finnish species samples used as reference panels. HI_paint values are based on 79,336 ancestry-informative markers displaying allele frequency differences $\geq$ 80% between non-Finnish species samples. (XLSX)

**S2 Table. Demographic parameters estimated by fastsimcoal2 in demographic model analyses.** Unless bounded, the upper limit of the search range could be exceeded. Each model used only a subset of these parameters. The time of admixture parameter (TADMS) indicated with an asterisk (*) was first unconstrained, and then constrained to test for recent hybridization (<50 generations). Double asterisks (**) mark parameters that calculation changes between models. The alternative minimum and maximum bounds are displayed in the respective columns. (XLSX)

**S3 Table. Maximum likelihood parameter estimates for all models concerning the history of the *F. aquilonia* × *F. polyctena* hybrid population sampled in Pikkala (contained 348,228 sites).** All effective sizes ($N_e$) are given in number of haploids. Times are given in number of generations. Migration rates are scaled according to population effective sizes (2Nm). Maximum-likelihood estimates for parameters are taken from the run reaching the highest composite likelihood of the 100 runs performed. Likelihoods are given in logarithmic scale. Maximum observed likelihood for this dataset is −1,377,703.973. ΔLikelihood is calculated by subtracting the expected likelihood from the maximum observed likelihood. (XLSX)

**S4 Table. Maximum-likelihood parameter estimates for all models concerning the history of the *F. aquilonia* × *F. polyctena* hybrid population sampled in Bunkkeri (contained 463,401 sites).** All effective sizes ($N_e$) are given in number of haploids. Times are given in number of generations. Migration rates are scaled according to population effective sizes (2Nm). Maximum-likelihood estimates for parameters are taken from the run reaching the highest composite likelihood of the 100 runs performed. Likelihoods are given in logarithmic scale. Maximum observed likelihood for this dataset is −1,821,814.189. ΔLikelihood is calculated by subtracting the expected likelihood from the maximum observed likelihood. (XLSX)

**S5 Table. Maximum likelihood parameter estimates for all models concerning the history of the *F. aquilonia* × *F. polyctena* hybrid population sampled in LångholmenW (contained**

**289,948 sites).** All effective sizes ($N_e$) are given in number of haploids. Times are given in number of generations. Migration rates are scaled according to population effective sizes (2Nm). Maximum-likelihood estimates for parameters are taken from the run reaching the highest composite likelihood of the 100 runs performed. Likelihoods are given in logarithmic scale. Maximum observed likelihood for this dataset is −1,144,658.203. ΔLikelihood is calculated by subtracting the expected likelihood from the maximum observed likelihood.
(XLSX)

**S6 Table. Maximum-likelihood parameter estimates for all models concerning the history of the *F. aquilonia* × *F. polyctena* hybrid population sampled in LångholmenR (contained 223,927 sites).** All effective sizes ($N_e$) are given in number of haploids. Times are given in number of generations. Migration rates are scaled according to population effective sizes (2Nm). Maximum-likelihood estimates for parameters are taken from the run reaching the highest composite likelihood of the 100 runs performed. Likelihoods are given in logarithmic scale. Maximum observed likelihood for this dataset is −883,471.568. ΔLikelihood is calculated by subtracting the expected likelihood from the maximum observed likelihood.
(XLSX)

**S7 Table. Demographic parameters estimated by fastsimcoal2 in demographic model analyses.** Unless bounded, the upper limit of the search range could be exceeded. Each model used only a subset of these parameters.
(XLSX)

**S8 Table. Maximum likelihood parameter estimates for the models concerning the history of the *F. aquilonia* × *F. polyctena* hybrid populations sampled in Bunkkeri and Pikkala (contained 328,913 sites).** All effective sizes ($N_e$) are given in number of haploids. Times are given in number of generations. Migration rates are scaled according to population effective sizes (2Nm). Maximum-likelihood estimates for parameters are taken from the run reaching the highest composite likelihood of the 100 runs performed. Likelihoods are given in logarithmic scale. Maximum observed likelihood for this dataset is −1,509,459.108. ΔLikelihood is calculated by subtracting the expected likelihood from the maximum observed likelihood.
(XLSX)

**S9 Table. Maximum likelihood parameter estimates for all models concerning the history of the *F. aquilonia* × *F. polyctena* hybrid population sampled in Bunkkeri and LångholmenW (contained 282,215 sites).** All effective sizes ($N_e$) are given in number of haploids. Times are given in number of generations. Migration rates are scaled according to population effective sizes (2Nm). Maximum-likelihood estimates for parameters are taken from the run reaching the highest composite likelihood of the 100 runs performed. Likelihoods are given in logarithmic scale. Maximum observed likelihood for this dataset is −1,291,308.181. ΔLikelihood is calculated by subtracting the expected likelihood from the maximum observed likelihood.
(XLSX)

**S10 Table. Maximum likelihood parameter estimates for all models concerning the history of the *F. aquilonia* × *F. polyctena* hybrid population sampled in Pikkala and LångholmenW (contained 218,545 sites).** All effective sizes ($N_e$) are given in number of haploids. Times are given in number of generations. Migration rates are scaled according to population effective sizes (2Nm). Maximum-likelihood estimates for parameters are taken from the run reaching the highest composite likelihood of the 100 runs performed. Likelihoods are given in logarithmic scale. Maximum observed likelihood for this dataset is −1,001,623.462. ΔLikelihood is

calculated by subtracting the expected likelihood from the maximum observed likelihood.
(XLSX)

**S11 Table. Maximum-likelihood parameter estimates for all models concerning the history of the *F. aquilonia* × *F. polyctena* hybrid population sampled in LångholmenW and LångholmenR (contained 150,207 sites).** All effective sizes ($N_e$) are given in number of haploids. Times are given in number of generations. Migration rates are scaled according to population effective sizes (2Nm). Maximum-likelihood estimates for parameters are taken from the run reaching the highest composite likelihood of the 100 runs performed. Likelihoods are given in logarithmic scale. Maximum observed likelihood for this dataset is −693,262.061. ΔLikelihood is calculated by subtracting the expected likelihood from the maximum observed likelihood.
(XLSX)

**S12 Table. Observed and simulated average levels of diversity (π) for each population in each comparison.** Simulated estimates were generated with msprime using both models inferred with fastsimcoal2 (SO: single origin scenario, IO: independent origin scenario). Simulated estimates were averaged over 100 independent runs.
(XLSX)

**S13 Table. Observed and simulated average levels of divergence ($F_{ST}$) between populations within each comparison.** Simulated estimates were generated with msprime using both models inferred with fastsimcoal2 (SO: single origin scenario, IO: independent origin scenario). Simulated estimates were averaged over 100 independent runs.
(XLSX)

## Acknowledgments

We thank G. Barroso for assistance with iSMC, the SpecIAnt group for feedback, and CSC–IT Center for Science, Finland, for computational resources. This work was performed under the Global Ant Genomics Alliance.

## Author Contributions

**Conceptualization:** Pierre Nouhaud, Simon H. Martin, Jonna Kulmuni.

**Funding acquisition:** Jonna Kulmuni.

**Investigation:** Pierre Nouhaud, Beatriz Portinha.

**Methodology:** Pierre Nouhaud, Simon H. Martin, Beatriz Portinha, Vitor C. Sousa, Jonna Kulmuni.

**Project administration:** Pierre Nouhaud, Jonna Kulmuni.

**Resources:** Pierre Nouhaud, Simon H. Martin, Vitor C. Sousa, Jonna Kulmuni.

**Supervision:** Pierre Nouhaud, Jonna Kulmuni.

**Visualization:** Pierre Nouhaud.

**Writing – original draft:** Pierre Nouhaud, Simon H. Martin, Jonna Kulmuni.

**Writing – review & editing:** Pierre Nouhaud, Simon H. Martin, Beatriz Portinha, Vitor C. Sousa, Jonna Kulmuni.

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
