## [Editor Report · Decision Letter 0]

17 Mar 2022

Dear Dr Nouhaud, 

Thank you for submitting your manuscript entitled "Rapid and repeatable genome evolution across three hybrid ant populations" for consideration as a Short Reports by PLOS Biology.

Your manuscript has now been evaluated by the PLOS Biology editorial staff, as well as by an academic editor with relevant expertise, and I'm writing to let you know that we would like to send your submission out for external peer review.

Once your full submission is complete, your paper will undergo a series of checks in preparation for peer review. Once your manuscript has passed the checks it will be sent out for review. To provide the metadata for your submission, please Login to Editorial Manager (https://www.editorialmanager.com/pbiology) within two working days, i.e. by Mar 21 2022 11:59PM.

If your manuscript has been previously reviewed at another journal, PLOS Biology is willing to work with those reviews in order to avoid re-starting the process. Submission of the previous reviews is entirely optional and our ability to use them effectively will depend on the willingness of the previous journal to confirm the content of the reports and share the reviewer identities. Please note that we reserve the right to invite additional reviewers if we consider that additional/independent reviewers are needed, although we aim to avoid this as far as possible. In our experience, working with previous reviews does save time. 

If you would like to send previous reviewer reports to us, please email me at rroberts@plos.org to let me know, including the name of the previous journal and the manuscript ID the study was given, as well as attaching a point-by-point response to reviewers that details how you have or plan to address the reviewers' concerns. 

Given the disruptions resulting from the ongoing COVID-19 pandemic, please expect some delays in the editorial process. We apologise in advance for any inconvenience caused and will do our best to minimize impact as far as possible.

Kind regards,

Roli Roberts

Roland Roberts

Senior Editor

PLOS Biology

rroberts@plos.org

---

## [Decision Letter · Decision Letter 1]

20 May 2022

Dear Dr Nouhaud,

Thank you for your patience while your manuscript "Rapid and repeatable genome evolution across three hybrid ant populations" was peer-reviewed at PLOS Biology. It has now been evaluated by the PLOS Biology editors, an Academic Editor with relevant expertise, and by four independent reviewers. 

You'll see that while the reviewers are broadly positive about the study, they each raise a number of concerns; strikingly, for example, several of the reviewers think that your evidence for the hybrid populations being truly independent is currently inadequate, and this key issue must be addressed before further consideration.

In light of the reviews, which you will find at the end of this email, we would like to invite you to revise the work to thoroughly address the reviewers' reports.

Given the extent of revision needed, we cannot make a decision about publication until we have seen the revised manuscript and your response to the reviewers' comments. Your revised manuscript is likely to be sent for further evaluation by all or a subset of the reviewers.

**IMPORTANT - SUBMITTING YOUR REVISION**

*Re-submission Checklist*

*Published Peer Review*

*PLOS Data Policy*

*Blot and Gel Data Policy*

Sincerely,

Roli

Roland Roberts

Senior Editor

PLOS Biology

rroberts@plos.org

REVIEWERS' COMMENTS:

Reviewer #1: 

I enjoyed reading the manuscript, which was well written and addresses a timely question about hybridization using approaches from an evolutionary genomics perspective. The main results are that (probably) independent hybrid populations (i) show similar evolutionary sorting of ancestry, and (ii) that the tendancy for regions to be sorted is predictable based on features such as gene density, previous history of selection, and local recombination rate.

The system is compelling, the data are high quality and this quality is matched by a quality analysis. This work will clearly be the foundation for a high-quality and long-term research programme addressing the most outstanding questions in the field.

A side effect of timeliness, however, is a rapidly expanding literature among which this work will stand. Importantly, I found that this paper did not cite several recent and highly relevant articles (I am not an author on any papers referenced). In particular, recent work in swordtail fish, most importantly including Langdon et al. 2022 PLos Genetics, should probably be referenced and discussed. I would also like to see the patterns put more into the context of emerging theory, for example by Carl Veller (bioRxiv 2019 and ensuing articles). 

I have minor concerns about the analysis that can probably be solved with a bit more transparency. In particular I am slightly concerned that hybrid incompatibilities could be filtered out by the Hardy-Weinberg based heterozygosity filter. I'd also like to see a bit more empirical rigour about why the authors suppose the mitochondria are such good evidence for independence. 

Finally, the authors do not really discuss or address hybrid incompatibilities outside of the first paragraph of the main text. As BDMIs are hypothesized to be a major driver of hybrid genome evolution, this seems like a rather large omission.

Line-by-line comments:

Line 18: This is good framing for the study, but I believe this is becoming increasingly clear. 

Line 38: I would argue that an additional reason to study is that it is a way to 'kick the tires' of the genome, to see how it works. We can discover new gene functions and interactions sometimes only in hybrids.

Line 65: This seems like a bit of a bait-and-switch when the models are equally supported below.

Line 66: It would seem that a reference to Molly Schumer's work (published in PNAS) where assortative mating maintains two separate sympatric hybrid populations would be appropriate here.

Line 66-9: It might seem trivial, but I think referencing a paper or supplementary result showing this explicitly would be valuable, especially in light of results below that suggest this is equivocal. This is important because the extent to which the paper's findings are of interest is largely determined by the degree of independence.

Line 71: Seems to contrast with statement below stating that F. polcytena ancestry is "more prevalent genome-wide".

Line 75: Unclear from the text whether the models also included mitochondrial haplotype.

Line 83: This paragraph is well-written and the analyses are strong.

Line 100: Could migration of haploid males (who would not transmit mitochondria) perhaps be responsible? (Am not an ant expert so this might be an ignorant statement).

Line 159: Suggest specifying whether sampled populations were taken from single-species colonies in areas of allopatry or in areas of sympatry (i.e., locations are given but this does not necessarily imply that they are allopatric).

Line 193: Recent theory and empirical studies suggest that hybrid incompatibilities cause selection for positive heterozygosity (Simon et al. 2018, Evol Letters; Thompson et al. 2022, PLoS Biology). Moreover, selection against recessive incompatibilities could also leave this signature. Could the authors comment on whether this step might filter out real signal? Do these sites show biased ancestry? What is the typical heterozygosity of excluded sites? Also, could the authors provide information and a graph to see if this was affecting regions or single SNPs? I'd like to know how much data loss this specific step resulted in. 

Line 226: Could these SNPs be used in the 'selection' analyses? 

Line 279: Are migrants haploid males, diploid females, or both? 

Figure 1: Panel C, K=2 is not what one expects from reading the text. The hybrid populations do appear to show minor parent ancestry and different directions for different populations. 

Figure 2: Panel A, not immediately clear why Scaffold 13 is used. Perhaps the figure could note why it was selected. Panels C-D, I must say that I found the 'topology weighting' approach from the figure was not what I expected from the text. The approaches taken by Langdon et al. 2022 PLoS Genetics are, to my eye, more intuitive and easier to evaluate, though I must say that after spending some time with the analysis I like the weighting approach and it is sound. I think the inset Figures added to my initial confusion: for the toplogies it appears as though the 'hybrid' is a separate entity from the two parent species, when the approach is really asking which of the two parents the sequences from hybrid populations resemble. Perhaps some annotation could help here, e.g., 'hybrid population sequence similar to F. polyctena'.

Figure 3: The presentation of data in panel A is wonderful. I haven't seen it before and if the authors came up with it, I think it is quite an innovation. For Panel B it would seem to be the axes should be flipped—isn't the statement that coding SNPs predicts ancestry, not that ancestry predicts coding SNPs?

Figure 4: This is also a very cool result and I think quite novel. In the caption instead of 'plain' I might suggest 'solid'.

Reviewer #2:

This paper provides evidence that sorting of ancestry in hybrid ant populations is repeatable and predictable based on inferred patterns of selection and recombination rate variation. The degree of inferred repeatability (correlations >0.5 for ancestry between pairs of populations) is really remarkable. If the results hold (see my comments below) this is a really exciting study. 

My main concern is that interpretation of the results depends heavily on whether (or rather to what extent) the four hybrid ant populations evolved independently (i.e., independent origins and subsequent connection versus not by gene flow). The authors are clearly aware of this as well, and several analyses are aimed at addressing the degree of independence. I appreciate this, but I am not yet fully convinced that the paper has sufficiently ruled out ancestry correlations being mostly driven by shared histories.

First, as the authors note, models of shared versus independent origins had similar likelihoods. This concern is somewhat tempered by the evidence that a modest number of generations (~20% of the time since admixture or 10 out of 50 generations) passed between admixture and population splitting in the single origin models. Still, it is not clear what proportion of the observed ancestry correlations can be explained by a single origin model with ~10 generations of shared history (I am not just interested in whether the correlations are higher than expected under such a null model, but how much higher). This is an important omission. 

Second, it is not clear that the best models from fastsimcoal2 are good models for the data. Specifically, the paper notes several ways in which these demographic models fail to predict patterns of ancestry (e.g., degree of sorting of ancestry). If the models do not explain the data particularly well (even if they are the best of the models considered) than it is harder to interpret deviations from predictions from the models as evidence of selection. I think two things should be done to address this issue. First, it would be nice to see that data simulated under the best models recreate patterns of genetic variation within and among the populations in general (even if not patterns of ancestry). Second, and especially if the models do not appear to predict genetic variation well, I think demographic models (especially models for single versus multiple origins) should be fit for the ancestry data directly (perhaps from the simple, naive chromosome painting, but any would be fine). It would be very interesting to see whether such models would suggest a single origin with more shared history.

My other more minor comments (with approximate line numbers) follow:

L35-45. I appreciate the brevity of the introductory paragraph but think that it consequently has fallen a bit short on putting the work in context. Most notably, this is not the first study to look at patterns genome sorting (genome stabilization) in hybrid lineages. Perhaps the best example concerns hybrid sunflowers (see, e.g., https://doi.org/10.1111/j.1558-5646.2007.00267.x; DOI: 10.1126/science.1086949), but other examples that consider repeated instances of hybrid lineage formation (Italian sparrows, Lycaeides butterflies) or that explicitly consider selection and recombination (swordtail fish) exist. This doesn't all need to be cited, but some reference somewhere to the relevant literature would be good. Likewise, the references in this section are all from the past few years but thinking on this topic goes back decades. Lastly, the term genome stabilization has been used in many papers to refer to what is here called genome sorting. I don't have a strong preference for one term over the other, but the connection should be made parenthetically.

L55-58. I think it is a bit strong to refer to this as "an ideal test case" when the evidence is only that the populations "may have independent origins" (an ideal case would be if there was very strong evidence or near certainty of independent origins).

L89. I recommend reporting the range of correlations here. They are high and worth calling out in the main text (just stating that the correlation is significantly different than 0 is much less interesting).

L93. Consider simulating not just from the best model, but from several of the best competing models.

L112-125. These patterns of enrichment are quite cool. They definitely increase my confidence that selection has contributed to the patterns of observed ancestry, though they still don't rule out a major role for shared history for the similarities among populations.

L179. State what species the reference genome is for here.

L184. Please define "normalized" in this context.

Reviewer #3:

This manuscript assesses the repeatability of genetic outcomes of hybridization in four Finnish populations of ants (in three localities), each with mixed ancestry from the widespread species Formica aquilonia and Formica polyctena. The authors find that each of the hybrid populations has fixed F. aquilonia ancestry in some parts of the genome and F. polyctena ancestry in others, and that the same genomic regions tend to be fixed for the same parental species ancestry across multiple hybrid populations . Additionally, genomic regions with low recombination rate and high gene density tend to fix alleles from the parent species with the larger inferred population size. Taken together, this suggests that natural selection shapes the similar outcomes of hybridization in multiple hybrid populations. The work addresses a consequential question in the field of speciation genetics , and the writing and figures are clear and easy to follow. 

There are two areas where I am not fully convinced: First, the independence of the three localities with hybrid populations seems a bit overstated as currently written. As the authors note, the likelihoods of models assuming a single origin of hybrids vs. independent origin of each hybrid population are similar. The main text doesn't point out that the likelihoods are higher for single origin than independent origin in all four pairwise comparisons of hybrid populations, suggesting that the histories of these Finnish hybrid populations are not completely independent of each other. The argument in the text that mitochondrial haplotype networks support three independent origins of hybridization doesn't appear very strong; there is substantial mitochondrial polymorphism in both parent species, and some of this variation could have been present in a single ancestral hybrid population before fixing different alleles in different descendant hybrid populations. 

A more substantial logical inconsistency is that the study relies heavily on analyses that assume neutrality in order to argue for pervasive effects of selection across the genome. To what extent and in what direction would the inferences of demographic history and recombination be biased by the selection required to produce the observed levels of sorting of parental variation in the hybrid populations, and the number and extent of inferred selective sweeps in the parent species? The demographic results, in particular, would be more convincing if similar results are obtained when restricting the analysis to genomic regions that are not sorted in hybrids, and/or have low [mu] scores in the scan for signatures of selection. In fact, convergent selection in hybrid populations with truly independent origins might even produce a false signal of shared ancestry or gene flow.

Minor details:

L386: change "was ran" to "was run"

Tables S3 and S4 have Excel formula errors, such that Δlikelihood and LogLikelihood are identical for four of the models in each table.

Reviewer #4:

This paper presents fascinating findings. Namely, that three separate hybridization events have occurred between two closely related species of ants, and that each hybrid population tends to share similar ancestry components along the genome. They suggest this pattern of correlated ancestry component sorting cannot be explained by chance and that shared selection pressure for specific ancestry components is the best explanation. I found the paper to be well-written and their approach and methods to be sensible and clearly justified. In particular, the authors take considerable efforts to model and account for demographic history and recombination rate variation, both of which effect signatures of selection and hybridization.

Despite very much liking the paper overall, I have some criticisms related to how the authors presented and used simulations to support their claims. 

In the results section on line 90, the authors state: "To test whether such predictability would be expected under neutrality, we used MSPRIME (15) to simulate neutral admixture scenarios following the best history and demographic parameters inferred for each population pair"

Related to this in the methods section on line 380 they state: "To do so, we used the parameters inferred by the best model identified for each population pair with FASTSIMCOAL2 and ran simulations in MSPRIME (v1.0.2, 15), modeling 100 non-recombining 10 kbp blocks with a recombination rate of 10-6 within blocks."

This is more or less all the details readers are given in the text about the msprime simulations, and it is difficult for me to tell what exactly was done. If I understand correctly, the authors fit models with FASTSIMCOAL2 that included all their populations. Did the msprime simulations exactly match the models inferred in FASTSIMCOAL2? Were all populations and their demographic changes and sizes included in these simulations, or were simulations done separately for different pairs? It seems quite important to me that the authors did the former in order to make apples to apples comparisons to their empirical results, but I cannot quite tell from these sentences if that is what they did, or if simpler simulations with two populations at a time were considered. Please better clarify what was done and justify these decisions in the text. 

Second and more importantly, given the similar fit of the shared and independent origins models (as explained on line 75), the authors need to assess the correlated sorting of ancestry from simulations for both model types. Does a model assuming shared origins have correlations close to the observed data, or does selection still need to be invoked to explain the observed patterns, even if the shared origins model is actually the correct one? Related to my comments above, using msprime simulations that include all populations simultaneously (rather than pairs) seems important to adequately represent a shared hybrid origins model. Demonstrating that the correlation in ancestry along the genome is due to selection sorting the ancestry components, even if the inferred shared hybridization event model is actually correct, would greatly aid the main (and most exiting) claims of the paper.

---

## [Decision Letter · Decision Letter 2]

27 Oct 2022

Dear Dr Nouhaud,

Thank you for your patience while we considered your revised manuscript "Rapid and predictable genome evolution across three hybrid ant populations" for publication as a Short Reports at PLOS Biology. This revised version of your manuscript has been evaluated by the PLOS Biology editors, the Academic Editor and three of the original reviewers.

Based on the reviews, we are likely to accept this manuscript for publication, provided you satisfactorily address the following data and other policy-related requests.

a) Please address my Data Policy requests below; specifically, we need you to supply the numerical values underlying Figs 1BCDEF, 2ABCDE, 3AB, 4AB, S1, S2, S3, S4, S5, either as a supplementary data file or as a permanent DOI’d deposition, e.g. part of your Figshare depo (if some of your Figures can be generated from the data already in Figshare, please clarify).

b) Please cite the location of the data clearly in all relevant main and supplementary Figure legends, e.g. “The data underlying this Figure can be found in S1 Data” or “The data underlying this Figure can be found in https://doi.org/10.6084/m9.figshare.c.6140793.v1”

We expect to receive your revised manuscript within two weeks. 

*Published Peer Review History*

*Press*

Sincerely,

Roli Roberts

Roland Roberts, PhD

Senior Editor,

rroberts@plos.org,

PLOS Biology

DATA POLICY:

Regardless of the method selected, please ensure that you provide the individual numerical values that underlie the summary data displayed in the following figure panels as they are essential for readers to assess your analysis and to reproduce it: Figs 1BCDEF, 2ABCDE, 3AB, 4AB, S1, S2, S3, S4, S5, NOTE: the numerical data provided should include all replicates AND the way in which the plotted mean and errors were derived (it should not present only the mean/average values).

SPECIES INDICATED IN THE ABSTRACT? 

- Please note that per journal policy, the model system/species studied should be clearly stated in the abstract of your manuscript. 

We require the original, uncropped and minimally adjusted images supporting all blot and gel results reported in an article's figures or Supporting Information files. We will require these files before a manuscript can be accepted so please prepare and upload them now. Please carefully read our guidelines for how to prepare and upload this data: https://journals.plos.org/plosbiology/s/figures#loc-blot-and-gel-reporting-requirements

DATA NOT SHOWN?

REVIEWERS' COMMENTS:

Reviewer #1:

I think the authors have done an exemplary job of engaging thoughtfully with reviewers. I am happy to sign off.

Reviewer #3:

The authors have done excellent work in response to my comments and, in my opinion, those of the other reviewers as well.

Reviewer #4:

I commend the authors for the considerable effort they put into addressing reviewer concerns. I think the manuscript is improved from their efforts. I do not have any further issues or recommendations to raise.

---

## [Editor Report · Decision Letter 3]

14 Nov 2022

Dear Dr Nouhaud,

Thank you for the submission of your revised Short Reports "Rapid and predictable genome evolution across three hybrid ant populations" for publication in PLOS Biology. On behalf of my colleagues and the Academic Editor, Leonie Moyle, I'm pleased to say that we can in principle accept your manuscript for publication, provided you address any remaining formatting and reporting issues. These will be detailed in an email you should receive within 2-3 business days from our colleagues in the journal operations team; no action is required from you until then. Please note that we will not be able to formally accept your manuscript and schedule it for publication until you have completed any requested changes.

Sincerely, 

Roli Roberts

Senior Editor

PLOS Biology

rroberts@plos.org